# Establishing 20S Proteasome Genetic, Translational and Post-Translational Status from Precious Biological and Patient Samples with Top-Down MS

**DOI:** 10.3390/cells12060844

**Published:** 2023-03-08

**Authors:** Angelique Sanchez Dafun, Dušan Živković, Stephen Adonai Leon-Icaza, Sophie Möller, Carine Froment, Delphine Bonnet, Adriana Almeida de Jesus, Laurent Alric, Muriel Quaranta-Nicaise, Audrey Ferrand, Céline Cougoule, Etienne Meunier, Odile Burlet-Schiltz, Frédéric Ebstein, Raphaela Goldbach-Mansky, Elke Krüger, Marie-Pierre Bousquet, Julien Marcoux

**Affiliations:** 1Institut de Pharmacologie et de Biologie Structurale (IPBS), Université de Toulouse, CNRS, Université Toulouse III—Paul Sabatier (UPS), 31077 Toulouse, France; 2Institute of Medical Biochemistry and Molecular Biology, University Medicine Greifswald, 17475 Greifswald, Germany; 3IRSD, Université de Toulouse, INSERM, INRAE, ENVT, Université de Toulouse III—Paul Sabatier (UPS), 31300 Toulouse, France; 4Internal Medicine Department of Digestive Disease, Rangueil Hospital, Université de Toulouse III—Paul Sabatier (UPS), 31400 Toulouse, France; 5Translational Autoinflammatory Diseases Section, LCIM, National Institute of Allergy and Infectious Diseases, National Institutes of Health, Bethesda, MD 20892, USA

**Keywords:** immunoproteasome, PRAAS, proteasome, proteoform, top-down proteomics, label-free quantification

## Abstract

The mammalian 20S catalytic core of the proteasome is made of 14 different subunits (α1-7 and β1-7) but exists as different subtypes depending on the cell type. In immune cells, for instance, constitutive catalytic proteasome subunits can be replaced by the so-called immuno-catalytic subunits, giving rise to the immunoproteasome. Proteasome activity is also altered by post-translational modifications (PTMs) and by genetic variants. Immunochemical methods are commonly used to investigate these PTMs whereby protein-tagging is necessary to monitor their effect on 20S assembly. Here, we present a new miniaturized workflow combining top-down and bottom-up mass spectrometry of immunopurified 20S proteasomes that analyze the proteasome assembly status as well as the full proteoform footprint, revealing PTMs, mutations, single nucleotide polymorphisms (SNPs) and induction of immune-subunits in different biological samples, including organoids, biopsies and B-lymphoblastoid cell lines derived from patients with proteasome-associated autoinflammatory syndromes (PRAAS). We emphasize the benefits of using top-down mass spectrometry in preserving the endogenous conformation of protein modifications, while enabling a rapid turnaround (1 h run) and ensuring high sensitivity (1–2 pmol) and demonstrate its capacity to semi-quantify constitutive and immune proteasome subunits.

## 1. Introduction

During evolution, the constitutive catalytic proteasome subunits (c20S) β1/β2/β5 gave rise to the catalytic immunoproteasome (i20S) subunits β1i/β2i/β5i [1]. Intermediate forms also exist, in which only one or two constitutive catalytic subunits are replaced [2]. The incorporation of these alternative immunoproteasome isoforms, which share ~60–70% sequence identity with their standard counterparts, modulates proteasome activity, regulator binding, substrate specificity and the amino acid sequence of the peptides generated [3,4,5,6,7]. We recently showed that substitutions occurring in the core particle subunits lead to conformational changes that alter the 20S interactome and function [8,9]. The i20S and its intermediates are constitutively expressed in immune cells, however, their expression can be induced in almost any tissue by proinflammatory stimuli [10]. In infected tissues, it can replace up to 90% of the c20S [11]. The peptides generated by the i20S may serve as epitopes, presented by Major Histocompatibility Complex (MHC) class I molecules at the cell surface. T lymphocytes use these MHC class I/peptide complexes to differentiate healthy from abnormal or infected cells. A peptide recruited into the MHC-class I complex must fulfill two prerequisites: an appropriate length of 8–9 amino acid residues and certain C-terminal anchor residues, which are basic or hydrophobic in humans. The replacement of standard subunits by immunosubunits alters the peptide profile generated by proteasomes: the resulting peptides have C-termini that are not acidic and more hydrophobic, resulting in a greater number of successfully presented antigens [12]. One role of the i20S is thus to modulate the MHC class I antigen presentation in cases of infection or transformation, thereby improving the immune response. Beyond antigen presentation, the i20S is essential in maintaining the proliferation of T cells in response to infection [13] and oxidative stress [14,15,16] or in maintaining protein homeostasis upon cytokine signaling or in immune cells [17,18,19,20].

Post-translational modifications (PTMs) or mutations in proteasome subunits and activators can alter proteasome activity. PTMs by phosphorylation, methylation, acetylation, ubiquitination and myristoylation have been linked to either increases or decreases in proteasome activity; however, their precise role in proteasome function still needs to be elucidated (reviewed in refs. [21,22]). Pathologies including cancers, heart diseases, neurological diseases, type 2 diabetes, inflammatory bowel disease (IBD) and proteasome-associated autoinflammatory syndromes (PRAAS), to name a few, have also been directly linked to the altered activity of the proteasome (reviewed in refs. [23,24,25,26]). PRAAS span the disease spectrum, ranging from chronic atypical neutrophilic dermatosis with lipodystrophy and elevated temperature (CANDLE) to joint contractures, muscle atrophy, microcytic anemia and panniculitis-induced lipodystrophy (JMP), Nakajo-Nishimura syndrome (NNS) and Japanese autoinflammatory syndrome with lipodystrophy (JASL), which are autoinflammatory syndromes mainly caused by genetic mutations in the immunoproteasome subunit β5i (encoded by *PSMB8*) [18,27,28,29,30,31,32,33]. Previous publications in PRAAS also reported mutations in β1i (*PSMB9*) [34], and β2i (*PSMB10*) [35]; compound heterozygous mutations in the constitutive subunit β7 (*PSMB4*) [18,36]; digenic mutations in β5i and constitutive subunit- α7 (*PSMA3*) or β7 (*PSMB4*), and in β1i and β7 [18]; and mutations in the 20S chaperones PAC2 and POMP [18,37,38]. More recently, mutations in the subunits of the 19S (the major 20S regulator) RPN5 (*PSMD12*) or RPT5 (*PSMC3*) have been linked to neurodevelopmental disorders [39,40,41].

The proteasome has also been the target for drug development. Three proteasome inhibitors targeting its chymotrypsin-like activity have been approved by the FDA since 2003 for the treatment of multiple myeloma [42,43,44]. However, a better understanding of proteasome regulation would allow more specific therapies targeting specific proteasome subtypes, thus avoiding unnecessary shutdown of downstream pathways that may cause off-target side-effects [43,44,45]. The group of Marcus Groettrup showed that specific inhibition of the i20S reduces graft rejection [46] and suppresses the progression of colorectal cancer in mice [47], but recent results have shown that targeting a single catalytic subunit of the i20S was not sufficient to achieve efficient therapeutic goals [45]. In this regard, quantifying the relative abundance of the i20S vs. c20S proteasome within a sample/cell extract or biopsy is of paramount importance.

To date, most groups working on proteasome function have assessed and investigated the i20S proteasome, PTMs and proteasome mutations by Western blot (WB) using specific antibodies [18,48,49]. Although very sensitive and compatible with complex samples (typically cell lysates), WB has limitations including cross-reactivity of certain antibodies, slower throughput, and poor quantitation capacities, and the method hardly provides valuable information regarding the nature and relative abundances of the detected proteoforms, which can include mature forms as well as truncated, proteolyzed, variant and/or modified isoforms. In addition, epitope tagging of mutant and wild-type (WT) constructs is often necessary to monitor PTMs and the impact of disease-causing mutations [18,49]. Our lab developed a series of mass spectrometry (MS)-based bottom-up (BU) proteomics methods aimed at detecting and quantifying proteasome subunits, including two-dimensional gel electrophoresis and isotope-coded affinity tag (ICAT) labeling [50], label-free proteomics [51,52], protein correlation profiling [5], stable isotope labeling by amino acids in cell culture (SILAC)-based multiplexed LC-selected reaction monitoring (SRM) [15], and more recently, top-down (TD) proteomics [8,9].

The latter consists of analyzing entire protein, rather than trypsin-digested proteins, as is commonly the case in BU proteomics, with the advantage of directly obtaining the accurate molecular weights (MW) of the proteins, and hence their potential combination(s) of PTMs, putative truncations and/or mutations and single nucleotide polymorphisms (SNPs) [53]. Furthermore, sequence coverage is usually pretty low in BU proteomics, and proteins are commonly identified and quantified based only on a small subset as low as one to three peptides; the BU database search only finds PTMs and mutations if these were previously included in database and searched for, and it is required that the peptides have the correct size and charge. The main current limitation of TD proteomics resides in the fact that proteins of different MW can have very different ionization yields, with smaller proteins usually displaying better “flyability”. It is commonly accepted that proteins <30 kDa are easily amenable to TD proteomics [54], which is ideal for the 20S proteasome, since all the α and β subunits have a MW between 20–30 kDa. The other advantage of TD proteomics is that the proteoform footprint can be relatively quickly obtained (~2 h including data analysis) and requires minimal amounts of proteins (~1–5 pmol), making it a convenient solution for quality control assessment and in-depth subunit characterization. We and others have already successfully applied this technique to 20S from commercial sources [8,55] or immunopurified from HEK cells expressing either the c20S or i20S [8]. More recently, we used TD proteomics to characterize the proteoforms of the spermatoproteasome immunopurified from bovine testes and muscle [9]. Beyond the identification and characterization of proteoforms, TD proteomics represents a novel avenue for protein quantification or semi-quantification (for a detailed review see [56]). Briefly, TD quantification methods are either based on metabolic labeling (i.e., SILAC [57]), isobaric chemical labeling (i.e., tandem mass tag [58,59]) or on label-free [54,60,61,62] approaches; each method has its pros and cons. Label-free quantification has the advantage of being less expensive and can be applied to any sample in theory (no need to use labelled media or feedstock), but multiplexing is not possible, and the method can arguably be less reproducible and sensitive. As mentioned before, differences in ionization yields can be challenging for label-free quantification, and this becomes even more evident in TD than in BU proteomics. Directly comparing the intensities of identical proteoforms in different samples/runs, an approach called “differential TD”, is a way to circumvent this problem [61].

In this work, we set out to develop a rapid TD-MS method to establish the proteoforms of 20S proteasome subunits and to gain information on their PTMs, mutations related to PRAAS and more common SNPs and immuno subunits on IFNγ-induced Caco2 cells, lung organoid and intestinal crypts. We also investigated the use of relative signal intensity for label-free quantification of the different subunits. Our approach consists of semi-quantifying the different 20S proteoforms within the same sample/run, which is quite different from differential TD. We first established relative ionization yields (RIY) for each standard and immunosubunit, using HEK EBNA cells producing either pure c20S or pure i20S [5,8]. We then applied this method to monitor the i20S induction by IFNγ in Caco2 cells and then to semi-quantify c20S vs. i20S in lung organoids and intestinal crypts.

## 2. Materials & Methods

### 2.1. Reagents

All reagents were provided by Sigma-Aldrich-Merck (St. Louis, MO, USA) unless otherwise specified. Substrates for activity assay (Suc-LLVY-AMC, Boc-LRR-AMC, and Z-LLE-AMC) were purchased from Enzo Life Science (Farmingdale, NY, USA). The protein quantity was estimated using the DC Protein Assay from Bio-Rad (Hercules, CA, USA).

### 2.2. Cell Lines, Organoids and Crypts

#### 2.2.1. HEK293-EBNA Cells

Human HEK293-EBNA cells expressing only either c20S or i20S, as previously described in ref. [2], were grown in Iscove’s Modified Dulbecco’s Medium (IMDM) supplemented with 10% fetal bovine serum (FBS), 116 mg/L L-Arginine and 36 mg/L L-Aspargine. For cells expressing i20S, the medium was also supplemented with 5 μg/mL puromycin and 600 μg/mL hygromycin B to maintain selective pressure. All cultures were supplemented with 100 units/ml penicillin, 100 μg/ml streptomycin, and maintained at 37 °C under 5% CO_2_. The medium was replaced every 2–3 days, and cell passaging was carried out at around 80% confluence. The cells were washed with Dulbecco’s phosphate-buffered saline (DPBS) three times and stored at −80 °C until use.

#### 2.2.2. Caco2 Cells

Caco2 cells were grown in Dulbecco’s Modified Eagle Medium (DMEM) Glutamax high glucose (4.5 g/L) plus pyruvate with 20% FBS. All cultures were supplemented with 100 units/ml penicillin, 100 μg/mL streptomycin, and maintained at 37 °C under 5% CO_2_. The medium was replaced every 2–3 days and cell passaging was carried out at around 70% confluence.

#### 2.2.3. B lymphoblastoid Cells

Immortalized B lymphoblastoid cell lines (BLCL) were generated from three PRAAS patients: A (β5i T75M homozygous), B (β5i T75M and α7 R233del), and C (β7 D212_V214del and β7 5’UTR c.-9G>A) as well as from the parents of Patient C, who are both clinically healthy but have β7 D212_V214del and β7 5’UTR c.-9G>A mutations, respectively. Cell lines were also generated from four other unrelated healthy controls. BLCL were cultured and amplified in RPMI 1640 medium supplemented with 10% FBS and 1% penicillin/streptomycin. Cells were washed twice with PBS and subsequently snap-frozen in liquid nitrogen.

#### 2.2.4. Airway Organoids

Healthy airway organoids were derived from human lung biopsies of healthy adjacent tissue from patients with lung cancer, as previously reported, and maintained in complete culture media [63,64]. On day 30 of culture, nine drops (each containing ~20 organoids) of 40 μL Cultrex growth factor reduced BME type 2 (Trevigen) previously seeded on a 24-well plate (Nunclon Delta surface, Thermo Scientific, San Jose, CA, USA) were washed twice with cold 1X phosphate-buffered saline (PBS). Each drop was harvested in 500 µL of 1X PBS and pooled in a 15 mL Falcon tube before centrifugation (800× *g* for 5 min). After centrifugation, the supernatant was discarded, and the pellet was resuspended in 2 mL of cold cell recovery solution (Corning) and kept at 4 °C for 20 min. Finally, organoids were centrifuged at 800× *g* for 5 min, and the pellet was washed twice with cold 1X PBS.

#### 2.2.5. Intestinal Crypts

Colonic samples were obtained from either surgical resections of a patient suffering from colorectal cancer or from a patient displaying a low-grade pseudoperitoneal myxoma of appendicular origin (Table 1). Crypts were harvested in healthy zones of the colon. Crypts extracted from the second patient were harvested at the margin of the resection, and at least 10 cm away from the tumor. 15,000 and 50,000 crypts were extracted from these patients, respectively, corresponding to ~1 mg and ~3 mg of total protein in their corresponding lysates, according to our protein DC assay. The patients’ characteristics were as follows:

### 2.3. Caco2 Cells Treatment with Different Proinflammatory Cytokines

In 6-well plates (Nunclon Delta surface, Thermo Scientific), Caco2 cells (seeding density: 8000 cells/cm^3^; in triplicates) were treated with different cytokines, namely 10 ng/mL interferon-γ (IFNγ), 100 ng/mL tumor necrosis factor-α (TNF-α), 100 ng/mL interleukin-6 (IL-6), 100 ng/ml interleukin-1β (IL-1β), or 10 µg/mL lipopolysaccharide (LPS), for three days. After washing the cells with DPBS three times, 0.5 mL of 5% SDS in 50 mM ammonium bicarbonate pH 7.6 was added per well, then the cells were scraped and transferred into 2 mL tubes. Tubes were stored at −80 °C until use.

### 2.4. Caco2 Cells Treatment with IFNγ

In a 150 mm Petri-dish (Nunc EasYDish, Thermo Scientific), Caco2 cells (seeding density: 8000 cells/cm^3^) were treated with 10 ng/ml IFNγ (PeproTech, Rocky Hill, NJ, USA) for 3, 7, and 11 days. The cells were washed with DPBS three times and stored at −80 °C until use.

### 2.5. Purification of c20S and i20S from HEK293-EBNA Cell Lines

Large scale purification of proteasomes was carried out using MCP21 antibody grafted in cyanogen bromide-activated sepharose beads, as described earlier [65]. Purification was carried out at 4 °C or in an ice bath. Cells were lysed with lysis buffer at pH 7.6 containing 20 mM Tris-HCl, 0.25% TritonX-100, 100 mM NaCl, 10 mM EDTA, and 1 tablet of protease and phosphatase inhibitors per 50 mL (cOmplete™ ULTRA Tablets Mini EDTA-fee and PhosSTOP, Roche, Basel, Switzerland) and sonicated with a Vibracell (10 cycles, 30 s on and 1 min off, 50% active cycle). The lysate was centrifuged at 16,000× *g* for 30 min, then the supernatant was filtered through a 0.22 µm membrane. The filtered lysate was then mixed with the beads grafted with MCP21 antibody and incubated overnight. The next day, the beads were washed with equilibration buffer (20 mM Tris HCl, 1 mM EDTA, 10% glycerol, 100 mM NaCl, pH 7.6) and eluted with equilibration buffer supplemented with 3 M NaCl. The eluate was concentrated down to 0.5 mL on centrifugal filters of 100 kDa MWCO and then separated using gel filtration chromatography. The chromatography was performed on a Superose 6 10/300 GL column, using TSDG buffer (10 mM Tris-HCl pH 7.0, 1 M KCl, 10 mM NaCl, 5.5 mM MgCl_2_, 0.1 mM EDTA, 1 mM DTT, 10% glycerol). The fractions were assayed by proteasome enzymatic activity and the active proteasome fractions (eluted fraction between 10–14 mL) were pooled together. Buffer exchange with equilibration buffer and concentration of fractions were done on centrifugal filters of 100 kDa MWCO. Additional glycerol was added to reach 20% and smaller aliquots were made. Aliquots were snap-frozen in liquid nitrogen and stored at −80 °C until use.

### 2.6. Purification of Proteasome from Caco2 Cells, BLCLs, Organoids and Crypts

MCP21 antibody (100 µg) was grafted and cross-linked (using 20 mM dimethyl pimelimidate in 0.2 M triethanolamine, pH 8.2) onto Protein G MagBeads (Genscript) (100 µL of 25% slurry) using the manufacturer’s protocol at room temperature. The purification was done at 4 °C or in an ice bath. Samples were lysed with lysis buffer at pH 7.6 containing 20 mM Tris HCl, 0.25% TritonX-100, 10% glycerol, 100 mM NaCl, 10 mM EDTA, 10 mM ATP, 5 mM MgCl_2_, and 1 tablet of protease and phosphatase inhibitors per 50 mL (cOmplete™ ULTRA Tablets Mini EDTA-free and PhosSTOP, Roche) and sonicated (Bioruptor Plus, Diagenode; 15 cycles, 45 s on and 15 s off) and then centrifuged at 16,000× *g* for 10–30 min. The supernatant (at least with 1–3 mg of total protein) was then mixed with the magbeads and incubated overnight. The next day, the beads were washed with equilibration buffer (20 mM Tris HCl, 1 mM EDTA, 10% glycerol, 100 mM NaCl, 2 mM ATP, 5 mM MgCl_2_, pH 7.6) and eluted with equilibration buffer supplemented with 3 M NaCl. The eluate was concentrated on centrifugal filters of 100 kDa MWCO while the buffer was exchanged with 200 mM ammonium acetate pH 7.4. Samples were analyzed directly to LC-MS, if possible. Otherwise, small aliquots of 10 μL were snap-frozen in liquid nitrogen and stored at −80 °C until use.

### 2.7. Detailed Top-Down LC-MS Analysis

The purified 20S aliquots were thawed and diluted (1:1) with 2% acetonitrile (ACN) and 0.02% formic acid (FA) in milliQ water. They were analyzed by online nanoLC using UltiMate 3000 RS autosampler and UltiMate 3000 RSLCnano LC system (ThermoScientific) coupled with an Orbitrap Fusion Tribrid mass spectrometer (Thermo Scientific) operating in positive mode through a Nanospray Flex^TM^ Ion source ES072 (Thermo Scientific). Each sample (1–2 pmol) was loaded onto a reverse-phase C4-precolumn (300 μm i.d. × 5 mm; Thermo Fisher Scientific) at 20 μL/min with loading buffer (milliQ H_2_O with 2% ACN and 0.2% FA). After 3 min of desalting, the precolumn was switched online with an analytical C4 nanocolumn (in-house packed Reprosil-PUR C4 3μm, 75 μm i.d. × 15 cm) equilibrated in 95% solvent A (milliQ H_2_O with 0.2% FA) and 5% solvent B (80% ACN with 0.2% FA and 20% milliQ H_2_O). Proteins were eluted using a binary gradient ranging from 5% to 40% (5 min), then 40% to 99% (33 min), and finally back to 5% (13 min) of solvent B at a flow rate of 300 nL/min. The MS was operated in Intact protein application mode with the Xcalibur 4.5 software (Thermo Fisher Scientific). The spray voltage was set to 1900 V, the ion transfer tube temperature to 350 °C, the RF lens to 60%, the normalized AGC target at 75% and the maximum injection time at 50 ms. For the MS-only method, scans were acquired in the 700 to 2000 m/z range with the Orbitrap resolution set to 15,000 and using 10 μscans for averaging. For the MS/MS method, the MS scans were acquired in the 1000 to 2000 m/z range with the resolution set to 7500 and using 10 μscans for averaging. The data-dependent mode was based on the cycle time, with 3 s between the master scans. The quadrupole isolation window was set at 5 m/z and electron-transfer/higher-energy collision dissociation (EThcD) fragmentation was used at 25% collision energy. Activation was achieved with ETD with a reaction time of 5 ms, a reagent target of 7 × 10^5^, and a maximum reagent injection time of 200 ms. The normalized AGC target was set at 400% and the maximum injection time at 400 ms. The MSMS scans were acquired on normal mass range with the resolution set to 60,000 and with 3 μscans.

The MS spectra were deconvoluted using RoWinPro software [55] and the results were visualized with VisioProtMS [66]. To determine the percentage of a particular proteoform, the sum of intensities from Rowinpro corresponding to each proteoform were attained and divided by the total sum of intensities corresponding to the subunit. In addition, MS/MS data were processed in Proteome Discoverer 2.3 (Thermo Scientific) to identify the proteoforms by sequence-matching the MS/MS fragments. A three-tier search was used for the processing workflow that contains an Absolute Mass Search with a narrow precursor mass tolerance of 100 Da, a BioMarker Search and another Absolute Mass Search with a large precursor mass tolerance of 1000 Da.

### 2.8. Label-Free Quantification in TD-MS using Relative Ionization Yields

The relative quantification was carried out using the intensity obtained after deconvolution of the raw data with UniChrom2 [67]. For each subunit, the intensities of known proteoforms were summed up. With the results coming from c20S or i20S purified from HEK-EBNA, the relative ionization yields (RIY) of each 20S subunit were calculated by dividing their intensity with the average intensity of all the non-catalytic subunits (Equation (1)).
(1)$$RIY_{Si}=\frac{A_{Si,}}{\frac{1}{n}\sum A_{Sncs}}$$
where *RIY_Si_* = relative ionization yield of a 20S subunit from pure c20S or i20S; *A_Si_* = abundance of a 20S subunit from pure c20S or i20S; *A_Sncs_* = abundance of non-catalytic 20S subunits (α1-7, β3,4,6,7) from pure c20S or i20S; and n = number of non-catalytic 20S subunits.

For the samples, the resulting 20S subunits’ intensities were corrected by dividing by their respective RIY, then normalized with the average intensity of non-catalytic subunits (Equations (2) and (3)).
(2)$$A_{Sx,corrected}=\frac{A_{Sx}}{RIY_{Si}}$$
(3)$$C_{Sx}=\frac{A_{Sx,corrected}}{\frac{1}{n}\sum A_{Sncs,corrected}}$$
where *A_Sx,corrected_* = corrected abundance of subunit x in the sample; *A_Sx_* = raw abundance of a subunit x in the sample; *A_Sncs,corrected_* = corrected abundance of non-catalytic 20S subunits (α1-7, β3,4,6,7) in the sample; and *C_Sx_* = relative amount of 20S subunit in the sample.

These results were compared with the calculated amount from just applying normalization with the average intensity of all the non-catalytic subunits (i.e., without the correction with RIY). Two sets of RIYs were established from three to four technical replicates of measurements with three injections of c20S and i20S in Tris and ammonium acetate buffers (the two buffers that we usually use for proteasome) for comparison. The relative quantification was first applied on different mixes of pre-purified c20S and i20S (c20S/i20S: 10%/90%, 20%/80%, 30%/70%, 50%/50%, 70%/30%, 80%/20% and 90%/10%) and then on endogenous samples such as immunopurified 20S from IFNγ-treated Caco2 cells, intestinal crypts and PRAAS-related samples. The proportions of c20S and i20S (which include all the immuno-containing 20S here) correspond to the relative abundance of β5 and β5i, respectively. Outliers were determined using the interquartile range method. Accuracy was calculated by finding the difference between the experimental and theoretical value divided by the theoretical value, while the coefficient of variation was the ratio of standard deviation to the mean in percentage.

### 2.9. Detailed Bottom-Up LC-MS Analysis

Lysates or eluents were supplemented with SDS to reach a final concentration of 5%, sonicated (15 cycles, 45 s ON and 15 s OFF at 10 °C) and centrifuged at 16,000× *g* for 10–30 min. Supernatant was then reduced with 100 mM tris(2-carboxyethyl)phosphine and alkylated with 400 mM 2-chloroacetamide at 95 °C for 5 min. Each sample was loaded on an S-trap spin column (Protifi, Fairport NY, USA), according to the manufacturer’s instructions, and digested with trypsin (Promega, Madison, WI, USA), corresponding to 2% of total protein weight (with a minimum of 1 μg), overnight at 37 °C. Digested peptide extracts were analyzed by online nanoLC using an UltiMate 3000 RSLCnano LC system (Thermo Scientific) coupled with an Orbitrap Fusion Tribrid mass spectrometer (Thermo Scientific) operating in positive mode. Each sample (1.0–2.5 µg peptides as analyzed by Pierce quantitative fluorometric peptide assay) was loaded onto a 300 mm ID 5 mm PepMap C18 pre-column (Thermo Scientific) at 20 ml/min in 2% ACN, 0.05% trifluoroacetic acid. After 5 min of desalting, peptides were on-line separated on a 75 mm ID 50 cm C18 column (in-house packed with Reprosil C18-AQ Pur 3 mm resin, Dr. Maisch; Proxeon Biosystems, Odense, Denmark) equilibrated in 95% buffer A (0.2% FA), with a gradient increased to 25% buffer B (80% ACN, 0.2% FA) for 75 min, then to 50% B for 30 min and then to 98% B for 10 min; they were held for 15 min before returning to starting conditions for 25 min, totaling an entire run time of 160 min at a flow rate of 300 nL/min. The instrument was operated in data-dependent acquisition mode using a top-speed approach (cycle time of 3 s). Survey scans MS were acquired in the Orbitrap over 400–1500 m/z with a resolution of 120,000, and a maximum injection time (IT) of 50 ms. The most intense ions (2+ to 7+) were selected at 1.6 m/z with quadrupole and fragmented by higher energy collisional dissociation (HCD). The monoisotopic precursor selection was turned on, the intensity threshold for fragmentation was set to 50,000, and the normalized collision energy (NCE) was set to 35%. The resulting fragments were analyzed in the Orbitrap with a resolution of 30,000. Dynamic exclusion was used within 60 s with a 10 ppm tolerance. The ion at 445.12003 m/z was used as the lock mass. The Mascot (Mascot server v2.8.1, Matrix Science, Boston, MA, USA; http://www.matrixscience.com (accessed on 3 June 2021)) database search engine was used for peptide and protein identification. Mass tolerance for MS and MS/MS was set at 10 ppm and 20 mmu, respectively. The enzyme selectivity was set to full trypsin with two missed cleavages allowed. The protein modifications were fixed carbamidomethylation of cysteines, variable oxidation of methionines, variable phosphorylation of serine, threonine and tyrosine, and variable acetylation of protein N-terminus. The UniProt proteome for both human and mouse was used as the database. The result files were then imported into Proline [68] for validation with false discovery rate (FDR) of ≤1.0% and label-free quantitation. Iterative alignment computation was applied using peptide identity with three as the maximum number of iterations. Alignment smoothing was carried out with a landmark range with 50 landmarks and 50% sliding window overlap. For the master map creation, the mapping tolerances were 5 ppm and 60 s for the m/z and time, respectively. For the statistics parameters, the *t*-test *p*-value of 0.01 was applied on both peptide and protein profile significant analysis. Only specific peptides were used, and the median ratio fitting was chosen as the abundance summarizer method. Normalization, missing values inference, *t*-test and z-test were also applied. IBAQ values were calculated by dividing the abundances with the number of observable peptides. The stoichiometry and relative amount of each 20S subunit, activators and interacting proteins were calculated using IBAQ values, as previously described by Fabre et al. [51]. For the treated Caco2 cells, mean and standard deviations were calculated from three biological replicates, while for the BLCLs, they were calculated from three injection replicates. Statistical difference was determined based on a one-way ANOVA with Tukey’s HSD test.

### 2.10. Proteasome Activity Assay

The activity assay was performed in 96-well black plates (Greiner Bio-One, Kremsmünster, Austria). A total of 10 µL of clear lysate (corresponding to 10 µg total proteins) was diluted with 40 µL of 100 mM Tris-HCl pH 8 buffer. Then, 50 µL of 100 µM fluorogenic peptide substrate, Suc-LLVY-AMC, Boc-LRR-AMC or z-LLE-AMC, to probe for chymotrypsin-, trypsin- or caspase-like activities, respectively, were added. The fluorescence intensities at λEx/λEm of 360/460 nm and at 37 °C were measured in a microplate fluorimeter (CLARIOstar, BMG Labtech, Ortenberg, Germany) for 12 cycles, with one reading every 5 min. The slope of the increase in fluorescence intensity over time (within linear range) was calculated, and normalization was done using the average slope for the controls. Mean and standard deviation were calculated from the three technical replicates. Statistical difference was determined based on a one-way ANOVA with Tukey’s HSD test.

## 3. Results

### 3.1. Establishing 20S Proteoform from Different Biological Materials

#### 3.1.1. Tri-Phosphorylated α7 (PSMA3) in Human 20S Identified by TD-MS

PTMs such as phosphorylation have been linked to either up- or down-regulation of proteasome activity [21,22]. In humans, phosphosites have been reported in 20S subunits including α2, α4, α7, β1 and β7, but the α7 phosphorylated form, which was found to regulate 26S stability [69] and Ecm29 binding [49], is by far the one that is detected in the widest variety of biological samples [8,22,55,70,71,72]. In order to gain better insight into human 20S proteoforms, we analyzed immunopurified 20S from HEK-EBNA cells expressing either c20S or i20S by TD-MS [2]. Importantly, as the immunopurification strategy used here targets α2, a subunit shared by all 20S proteasome subtypes, the 20S core particles analyzed in this work may correspond to in cellulo free, regulator-associated 20S proteasomes, or assembling 20S proteasomes [51]. In line with our previous work, we only observed phosphorylation of α7 [8,55]. For c20S, the acetylated *[2-255]* (28,344 Da) and mono-phosphorylated *[2-255]* (28,424 Da) forms that have been previously reported were both observed in MS (Figure 1A) and sequenced by MSMS (Figure 1B,C). Interestingly as shown in Figure 1A, additional proteoforms at 28,529 Da and 28,600 Da, were also observed. The proteoform at 28,529 Da exhibited a + 96 Da mass difference compared to the full-length α7 *[1-255]*, reflecting to a phosphorylation (+80 Da) and oxidation (+16 Da). On the other hand, the proteoform at 28,600 Da (one of the major proteoforms observed here) showed a + 298 Da mass excess compared with the 28,344 Da proteoform (acetylated α7 *[2-255]*), possibly corresponding to three phosphorylations and one oxidation (25 ppm mass error based on the MW of the precursor in MSMS analysis as shown in Figure 1D). The matched b- and c- ion fragments confirmed that it corresponded to α7; however, the lack of y- and z-ion fragments limited the localization of the additional phosphorylations and oxidation (Figure 1D). Note that the placement of the phosphorylation sites at S243 and S250 is based on BU proteomics literature [71,72,73,74], while phosphorylation at S142 is based on prediction by NetPhos3.1 (score 0.977) [75,76] (Appendix A) and the oxidation of the C-terminal methionine was previously reported by our group [77]. This suspected tri-phosphorylated form was not specific to the c20S since it was also observed in i20S as the main proteoform when it is freshly purified (Appendix A). However, it was found to be unstable upon storage, from 91% of total α7 after purification, down to 9% after three months at −80 °C (Figure 1E,F), which can be related to the previously observed truncations of α4 and α4s upon storage [9]. Furthermore, this putative tri-phosphorylated state was highly unstable under acidic conditions (0.2% formic acid, 2% acetonitrile), as its abundance decreased during the course of three consecutive injections (i.e., 92%, 44% and 10%), while the amount of the mono-phosphorylated proteoform increased (i.e., 1%, 32% and 69%), becoming the main α7 proteoform (Appendix A). These results indicate that human α7 is constitutively tri-phosphorylated but includes two highly labile phosphorylations, other than the previously described S250 phosphosite.

#### 3.1.2. Mutations in 20S Subunits in PRAAS Patients and Their Effect on 20S Assembly

PRAAS mutations have been identified in genes encoding 20S non-catalytic subunits such as α7 and β7, and immuno-catalytic subunits β5i, β1i and β2i [18,27,28,29,30,31,32,35]. Previous work used immunoblotting for tagged proteins and native-PAGE analysis to assess the impact of the mutation on 20S composition and assembly, a strategy that requires the long process of cell transfection and expansion as well as the challenging development of specific antibodies [18,35]. To quickly establish proteoform footprinting in endogenous samples and determine whether these mutations affect 20S assembly, we analyzed immunopurified 20S from BLCLs expanded from PRAAS patients and healthy controls with TD-MSMS, which is fast and has a greater specificity compared to antibody-based methods. We analyzed nine samples: three PRAAS patients, Patient A (β5i p.(T75M) homozygous), Patient B (digenic: β5i p.(T75M) and α7 p(R233del)), and Patient C (compound heterozygous β7 p.(D212_V214del) and β7 5’UTR c.-9G>A); the father and mother of Patient C who are both clinically healthy but have β7 D212_V214del and β7 5’UTR c.-9G>A mutations, respectively, and four other healthy controls.

Patient A had two proteoforms in the β5i subunit that, as expected, harbored the p.(T75M) mutation, one at 22,689 Da, that had a relative abundance of 90% and corresponds to the mature form *[73-276]*, and one at 22,705 Da with a relative abundance of 10%, which is its mature oxidized counterpart (Figure 2A). These two proteoforms were confirmed by their MWs (<1 Da mass accuracy) and MSMS sequencing (Figure 2B,C). These results confirm that (1) the mutation is homozygous, as no WT subunit with an expected MW of 22,659 Da was observed, and (2) the mutation does not impede the incorporation of the β5i subunit into the 20S in the course of assembly. However, despite proper incorporation, a significant decrease in chymotrypsin-like (*p*-value < 0.0001) but not trypsin-like and caspase-like activity was observed (Figure 2D and Appendix A), as was previously reported [27]. Additionally, we observed that the MW of β1i was 20 Da lower than that expected for the WT isoform (21,276 Da). By adding in our database all the 20S subunit variants present in UniProt, we identified the p.(R60H) SNP as the best fit for our MSMS data (Figure 2E). This highly prevalent allele is very common (allele frequency of 0.24, with 8575 homozygotes in the gnomAD database) and has no association with diseases, but would have probably been missed by a classical BU approach.

Complementary BU-MS analysis indicates a slight decrease (5 %) in the abundance of 20S in the lysate of this patient-derived BLCL compared to those from healthy donors (Appendix A). Label-free BU-MS analysis after 20S immunopurification reports a 1.5-fold increase in the amount of assembling 20S (based on the abundance of 20S-associated assembling chaperones, Appendix A) and no major difference in terms of co-immunopurified major 20S-associated proteins, apart from a threefold higher interaction with PA28αβ compared to healthy donors (Appendix A). Altogether, these results indicate that mutation of the catalytic p.(Thr75) mainly affects β5i activity with little impact on its incorporation, structure and subsequent 20S interactome.

Patient B harbors digenic mutations in β5i (p.(T75M)) and α7 (p.(R233del)), respectively, and the corresponding proteoform maps are shown in Figure 3A,B. In this case, both the mature WT (22,659 Da, 51% relative abundance) and mature p.(T75M) mutant β5i (22,689 Da, 41% relative abundance) were observed by TD-MS. Some partial oxidation was also observed for both the WT (22,675 Da, 6% relative abundance) and p.(T75M) mutant (22,705 Da, with 2% relative abundance). Both the WT and p.(T75M) proteoforms were confirmed by their MWs (<1 Da mass difference) and by TD-MSMS sequencing (Figure 3C,D and Appendix A). The relative abundance of the mutated (43 ± 3%) was slightly lower than for the WT (57 ± 3%), but clearly confirms that Patient B is heterozygous for the p.(T75M) mutation. Four α7 proteoforms were also observed, including the three main WT forms (acetylated at 28,344 Da, mono-phosphorylated at 28,424 Da, and tri-phosphorylated at 28,600 Da) and a lower abundant proteoform at 28,186 Da, which may correspond to the oxidized *[2-255]* form, with the expected deletion of p.(R233). Due to the low abundance of this p.(R233del) proteoform (4% of the total α7), it was not selected for MSMS sequencing. The only detected proteoform was the most abundant (79%) WT acetylated proteoform (Figure 3E). These results indicate that (1) Patient B is heterozygous for the T75M β5i mutation, since the WT proteoform was also observed in similar abundance to the T75M proteoform, (2) there is indeed an incorporation of the T75M β5i subunit in 20S complexes, albeit to a lower extent when compared to WT β5i, as previously suggested [18], and (3) the deletion of p.(R233) in α7 resulted in almost (4%) no incorporation of this proteoform in the 20S, as shown by Brehm et al. [18], thus resulting in reduced incorporation in the 20S (since 96 ± 1% of the signal corresponds to the WT form). A significant decrease in chymotrypsin- and trypsin-like activities was observed in the lysate from Patient B (Figure 3F and Appendix A); this is similar to what had already been reported [18]. The caspase-like activity was also decreased, albeit not significantly (Appendix A). The β1i p.(R60H) SNP was also found in Patient B, as confirmed by TD MSMS sequencing (Figure 3G).

Similarly to Patient A, our complementary BU-MS analysis indicated only a slight change (3 % decrease) in 20S abundance in the lysate of Patient B (Appendix A), whereas previous work reported a decrease in proteasome content due to reduced expression of the mutated α7 subunit [18]. BU-MS label-free analysis after 20S immunopurification also suggested a similar amount of assembling 20S (Appendix A) and no major change in interactions with the major 20S-associated proteins (Appendix A). Altogether, these results confirm that the mutation of the catalytic p.Thr75 affects mainly β5i activity, with little impact on its incorporation, structure and subsequent 20S interactome. Interestingly, the reduction in chymotrypsin-like activity for this heterozygous Patient B (Figure 3F), was ~half of that observed for the homozygous Patient A (Figure 2D). Furthermore, our results show that the R233del mutant of α7 is not incorporated in 20S proteasome (only by 4%). In order to discriminate between quantity and assembly defects, we next attempted to quantify the p.(R233del) vs. WT α7 in the lysate of Patient B, by semi-quantifying proteo-specific peptides. However, p.R233 is surrounded by two other trypsin cleavage sites (K/DIR/EEAEK/), meaning that the resulting peptide (DIR) is very small. Deletion of p.R233 removes a trypsin cleavage site and would generate the peptide DIEEAEK that could potentially be observed and quantified by our approach, but this is not the case for the very small DIR peptide present in the WT. However, the fact that no changes were detected in the levels of assembling 20S (Appendix A), suggests that there was no incorporation deficiency, but rather either a reduced protein expression or a decreased stability of this mutated α7.

For Patient C, harboring compound heterozygous mutations p.(D212_V214del) and 5’UTR c.-9G>A in subunit β7, the TD-MS proteoform map showed MWs corresponding to mature WT β7 *[46-264]* at 24,392 Da (main proteoform with 75% relative abundance) and its oxidized form at 24,408 Da (Figure 4A). These two proteoforms were confirmed by TD-MSMS sequencing with only <1 Da MW difference on the precursors (Figure 4B,C). When overlapping the β7 proteoform maps of Patient C with his parents (Figure 4D), we observed very similar results between Patient C and his mother, who carries the 5’UTR c.-9G>A SNP. In the case of the father with the p.(D212_V214del) mutation, β7 was observed at 24,380 Da and 24,396 Da, which are both 12 Da lower than the corresponding MWs in Patient C and his father. According to the variants present in the UniProt database, this 12 Da difference may correspond to the replacement of Ile234 by a Thr residue, which was further confirmed by TD-MSMS sequencing, in the father only (Figure 4E and Appendix A). This SNP has an allele frequency of 0.19, with 6086 homozygotes in the gnomAD database. Interestingly, with the complementary BU-MS results, both Patient C and his father had five to 15 times higher levels of the 20S-associated chaperones (PAC1-PAC2, and POMP), than the controls (Appendix A), which confirms previous data [18] and indicates that the p.(D212_V214del) mutation impedes β7 maturation, inducing a stalling of the assembly process. This is in accordance with the observed lower amounts of 20S proteasome (about a 15% decrease compared to controls) in the corresponding lysates (Appendix A) that is likely caused by the destabilization of the C-terminal tail of β7, which functions as a clamp between the two halves of the 20S proteasome complex [18,78]. Finally, unlike Patients A and B, Patient C shows only WT β1i. However, TD-MS analysis of his mother shows both WT and p.(R60H) proteoforms, which was also the case in two healthy patients (Appendix A). The results of Patient C and his parents thus show that (1) there is indeed no incorporation of the p.(D212_V214del)-mutated β7 subunit into the 20S complexes, as we only observed the WT forms; (2) the p.(D212_V214del) mutation on β7 results in a partial stalling of the 20S proteasome assembly process; and (3) the father of Patient C in addition to the p.(D212_V214del) also carries the p.(I234T) SNP on β7, which does not impede 20S incorporation and was not seen in Patient C.

#### 3.1.3. Detection of Immuno Catalytic Subunits in Different Complex Samples by TD-MS

The immuno subunits of the proteasome are abundantly expressed in immune cells and they can also be induced upon exposure to oxidative stress and proinflammatory stimuli. To monitor their expression and to test the sensitivity of our TD-MS workflow, we applied our pipeline to immunopurified 20S from samples of increasing complexity such as IFNγ-treated and non-treated Caco2 cells, healthy airway organoids, and intestinal crypts.

Before using TD proteomics, to see the effect of inflammation on the 20S proteoforms, we monitored the expression of the six catalytic subunits in the lysate by label-free BU quantification after treatment with interferon-γ (IFNγ), tumor necrosis factor-α (TNF-α), lipopolysaccharide (LPS), interleukin-6 (IL-6) and interleukin-1β (IL-1β). Comparison of the median of all the peptide intensities divided by the number of observable peptides (iBAQ [79]) of each subunit revealed that the c20S catalytic subunits were replaced by their i20S counterparts using 10 ng/mL of IFNγ for three days (Figure 5A). The other cytokine treatments failed to upregulate the i20S, as evidenced by PCR using IFNγ, TNFα and LPS [80].

We then treated Caco2 cells with 10 ng/ml IFNγ for 7 days to compare the proteasome TD-MS footprints before and after induction of the i20S. As expected, c20S was expressed constitutively in Caco2 cells and we observed a decrease in the signal of the standard catalytic subunits that were replaced by the i20S subunits upon treatment with IFNγ (Figure 5B).

In healthy airway organoids, β1i/β2i/β5i subunits were detected even without IFNγ induction (Figure 6A). These airway organoids are composed mainly of epithelial cells such as ciliated, club, basal and goblet cells, and i20S has already been identified in nonimmune lung cells under normal conditions [81,82]. Although the function of the i20S in these cells is not fully explained, its implication in bronchial epithelial cell differentiation has already been reported [83].

Next, we attempted to immunopurify 20S proteasomes from crypts derived from partial resection of the intestinal tract from two patients. In a first attempt, starting from ~15,000 crypts (~1 mg protein) from a healthy donor allowed us to characterize only 12 out of 17 proteasome subunits (Figure 6B), thus demonstrating the limits of detection with this method. We repeated the experiment, starting from ~50,000 crypts (~3 mg) from a patient with suspicion of colorectal cancer, which enabled us to generate a full proteoform map containing both the c20S and i20S catalytic subunits (Figure 6C). Note that β5 was not shown, as its intensity was below the threshold set for visualization. Visual comparison of the intensities of the subunits showed that this sample mostly contained the i20S. Overall, these results indicate that our TD-MS workflow is sufficiently sensitive to monitor the incorporation of the i20S, not only in biological models such as cells and organoids, but also in biopsy samples.

### 3.2. Relative TD Label-Free Quantification of c20S and i20S in Complex Mixtures

As mentioned previously, current quantitative TD-MS approaches do not take into account ionization yields, because they either compare two different proteoforms of the same gene and assume that they have similar ionization yield, or perform differential analysis by comparing the signal intensity of each proteoform in various samples (after signal normalization). Here, we set out to quantify the relative abundance of each 20S subunit within a single acquisition. To this end, we first established the relative ionization yield (RIY) of each c20S and i20S subunit by spraying pure c20S and i20S and different mixes of known composition (see Section 2 and Appendix A).

In order to further benchmark our approach, we compared the label-free quantification obtained by BU-MS and by TD-MS with and without application of the previously established RIY (Appendix A). In all tested cases (Caco2 cells, intestinal crypts and PRAAS patients), the application of the RIY allowed us to obtain results that were in better agreement with the label-free BU-MS quantification, but without the need to perform trypsin digestion, and with the additional information from the proteoform footprint (see Appendix A).

We thus detected the following using this label-free TD-MS quantification with RIY correction: 85% ± 7% i20S in Caco2 cells after 11 days of IFNy treatment, 90% ± 9% i20S in the intestinal crypt sample from normal tissue of a patient with suspicion of cancer, and an average of 80% ± 11% i20S in all the BLCL derived from healthy and diseased patients (Figure 7 and Appendix A).

These results indicate that (1) relative quantification of β5/β5i signal intensities in TD-MS can be used to estimate the c20S and immuno-containing 20S levels in endogenous samples; and (2) the application of RIY correction on raw abundances is required to determine a better estimate than one achieved by just performing the usual normalization.

## 4. Discussion

Our studies reported in this manuscript showed the advantages and current limitations of 20S immunopurification followed by TD-MS to unravel and semi-quantify its different proteoforms including PTMs, proteasome subunit mutations/SNPs and incorporation of i20S catalytic subunits.

In 20S proteasome immunopurified from HEK cells, Caco2 cells, BLCL from PRAAS patients and controls and lung organoids, we found a proteoform that matched the MW of triphosphorylated and oxidized α7 (Appendix A). The fact that this proteoform was found in HEK EBNA cells containing only i20S, and that a major increase in i20S upon activation of Caco2 cells with IFNγ was observed without changes in the relative abundance of mono vs. triphosphorylated α7, demonstrate that the presence of this proteoform does not correlate with the nature of the catalytic subunits. The two C-terminal serine residues (S243 and S250) were previously shown to be phosphorylated by protein kinase II [71] and have been confirmed in various proteomics and phosphoproteomics studies [69,70,72,73]. We here propose that the third phosphorylation might correspond to another site predicted on p.S142, as suggested by the NetPhos3.1 server. Early work from Castaño et al. [71], showed that deletion of the C-terminal part of α7 completely abolished the phosphorylation by protein kinase II, but has not been excluded that this third site might involve another kinase. In previous studies, triphosphorylated α7 has been reported in yeast, but two out of the three phosphosites were found on a protein segment that is absent in humans [49,55]. We found that one of these three phosphosites (probably S142) was particularly unstable upon freeze–thaw cycles, and even at 4 °C for a couple of hours, which may explain the absence of the phosphorylated residue in 20S from intestinal crypts (Appendix A). The crypts were kept at −80 °C for several months after immunopurification, which might have caused the loss of the unstable phosphorylation. The 3rd phosphosite has not been detected in classical proteomics/phosphoproteomics approaches and immunoblotting with phospho-specific antibodies that have long processing times and are thus prone to promote dephosphorylation of labile sites. We show here that the 3rd phosphorylation is completely lost in a couple of hours, which is compatible with our TD-MS workflow but not with overnight trypsin digestion (proteomics) or Western blot analysis. We will further investigate the function of this third phosphosite in the future, as phosphorylation of α7 was previously shown to regulate 26S stability [69] and Ecm29 binding [49]. Of note, any other PTM can be in theory identified by TD-MS, provided that it is stable in the gas phase, as we previously showed on the mycoloylation of porines from *Corinebacterium glutamicum* [84] or on the combined acylation and glycosylation of the LpqH antigen from *Mycobacterium tuberculosis* [85]. Concerning the ubiquitin proteasome system, detecting single or multiple O-GlcNAc [86,87,88], glutathionylation [89], ADP-ribosylation [90], succinylation [91] or myristoylation [92] could be of particular interest.

Our workflow combining TD-MS and BU-MS (Figure 8) allowed us to confirm mutations in BLCLs derived from PRAAS related patients and their parents and we provided further insights on the possible effect of these disease-causing mutations on 20S assembly. Our results confirmed that Patient A was homozygous for the p.(T75M) mutation in β5i and that this mutation mainly affected the chymotrypsin-like activity and only minimally decreased the incorporation and the binding of other molecules to the 20S. This is to be expected, as this mutation affects the third threonine from the catalytic triad of the β5i subunit conferring the chymotryptic-like activity. Previous modeling of this mutation predicted only a slight local structural rearrangement [18,27].

The proteoform footprint and the label-free TD-MS of the proteoforms of immunopurified 20S from Patient B showed very similar abundances of WT and p.(T75M) β5i, but almost no incorporation of p.(R233del) α7, confirming in a single experiment that only the first mutated isoform can be found in fully assembled 20S complexes. Work from Brehm et al. had demonstrated that this patient had normal α7 mRNA levels but no protein expression, thus suggesting that the p.(R233del) mutation affects exclusively the α7 stability as it probably alters subunit folding [18]. Although label-free BU proteomics was not able to distinguish and thus quantify the WT and R233del isoforms of α7, contrary to our label-free TD quantification, BU analyses showed no changes in terms of 20S content in the lysates and assembling 20S in the immunopurified complex, respectively, confirming this finding. Moreover, since the caspase-like activity of 20S from Patient A (homozygous for p.(T75M) β5i) and from a control harboring the p.(R60H) β1i SNP were not significantly affected, one can reasonably argue that the decrease in caspase-like activity observed in Patient B was solely due to this p.(R233del) in α7, even though synergistic effects between p.(R233del) α7 and p.(R60H) β1i cannot be excluded at this stage.

The effect of β7 p.(D212_214del) was even more severe than the α7 p.(R233del) (4% incorporation) as it could not be detected at all in the immunopurified 20S proteoform footprint. This was expected, since, as described previously, β7 is the last subunit being incorporated into the 20S before dimerization [93,94] and these three residues are located in a helix near its the C-terminal region, which has been reported to be essential for the dimerization step during assembly [18,78]. We could also confirm by TD-MSMS sequencing two SNPs that were also present in genetic analyses of the patients: p.(I234T) in β7 in the father of Patient C and p.(R60H) in β1i of Patients A and B (homozygous), in the mother of Patient C (heterozygous) and in two healthy controls. While the p.(R60H) SNP is very common and not disease-related, the p.(I234T) variant has been predicted to be one of the SNPs that has significant genotypic association with depression [95].

These patient-derived BLCL were instrumental to optimize our pipeline before applying it to more rare and precious samples such as biopsies. They showed the complementarity of the information obtained by both TD and BU approaches, as well as their limitations.

We then investigated whether our TD-MS workflow was able to monitor the 20S immuno subunits after immunopurification of lysates of increasing complexity, i.e., cell lines, organoids, and biopsies. We found out that a starting quantity of 1–3 mg total protein was enough to successfully establish these 20S proteoform footprints, implying that this method can be used for precious biological samples such as disease-related biopsies.

We also investigated its potential application for intra-acquisition relative quantification of different subunits, in order to quickly establish the proportion of c20S and i20S in a sample. We confirmed that the ionization yields of the different α and β subunits were quite different (ranging from 30% to 160% relative to the average of all the non-catalytic subunits), but reproducible. This allowed us to semi-quantify the relative amount of catalytic subunits after induction of the i20S in Caco2 cells and also in lung organoïds and colorectal crypts from patients. Even though the standard deviations were not negligible, the obtained values were in good agreement with the expected ratios (Appendix A), showing that this relative quantification with the correction of abundances with RIY using top-down MS can be used to determine the relative abundance of c20S vs. i20S proteasomes. Interestingly, β5 and β5i both show good ionization yields compared to the other catalytic subunits, and since they are present in all the immuno subunit-containing proteasomes (i.e., even the intermediates); quantifying the relative signal of β5 vs. β5i represents a fast and robust way of measuring the relative abundance of c20S vs. i20S, even for low-abundance sample in which the other catalytic subunits might not be detected.

## 5. Conclusions and Perspectives

The advantage of our workflow is threefold: it provides not only a relatively fast turnaround (approximately 2 h, including data analysis) but also a very thorough characterization of the different 20S proteoforms present in the sample; this is an aspect that is often overlooked by other methods. Finally, this very sensitive method is compatible with valuable and low-abundance clinical samples to detect the expression of variants or SNPs. For example, we identified the presence of multiple phosphorylated forms of α7, whose function(s) and precise localization, however, still need to be investigated. On the other hand, even though the method still suffers from inaccuracy due to reproducibility issues in terms of ionization yields, it allowed us to provide a good estimate of the 20S composition. In terms of sensitivity, we were able to analyze immunopurified 20S starting from 10 × 10^6^ Caco2 cells and from ~50,000 crypts, corresponding to ~1–3 mg of starting material in the lysate and ~1–2 pmol (0.7–1.4 µg) of 20S injected in the mass spectrometer; this is in good agreement with the quantity of total protein commonly injected on this instrument for shotgun BU experiments. The main bottleneck in terms of TD sensitivity compared to BU thus does not arise from the sensitivity of the mass spectrometer, but rather from its difficulty in analyzing complex mixtures (lysates), hence the need to immunopurify the 20S proteasome and start from higher quantity of material. However, the level of information obtained here is much higher, and we consider that the two approaches are complementary. Furthermore, as instruments are becoming not only more sensitive but also able to analyze more complex mixtures, we foresee that the use of TD proteomics will be exponential in the near future. For example, we expect that new developments such as proton charge transfer reduction [96] or charge detection mass spectrometry [97] combined with more efficient fragmentation techniques such as ultraviolet photodissociation [98], electron capture dissociation [99] or surface induced dissociation [100] will allow the analysis of higher MW and lower-abundance species. In the case of proteasome complexes, this will potentially expand our research to to co-immunopurified regulators or proteasome interacting proteins, since other diseases are related to mutations of chaperone (PAC2, POMP) or 19S subunits [18,37,41].

Now that we have showed that the workflow can be used in tissues/biopsies from patients, we intend to apply it to other proteasome subtypes or in different pathological contexts, for example, to investigate the role of the i20S in different inflammatory bowel diseases from resections of patients. Finally, the determination of RIY for intra-acquisition/semi-quantification of protein complex subunits can be applied to any other multiprotein/multiproteoform complexes, including ribosomes and inflammasomes, to name but a few.

## Figures and Tables

**Figure 1 cells-12-00844-f001:**
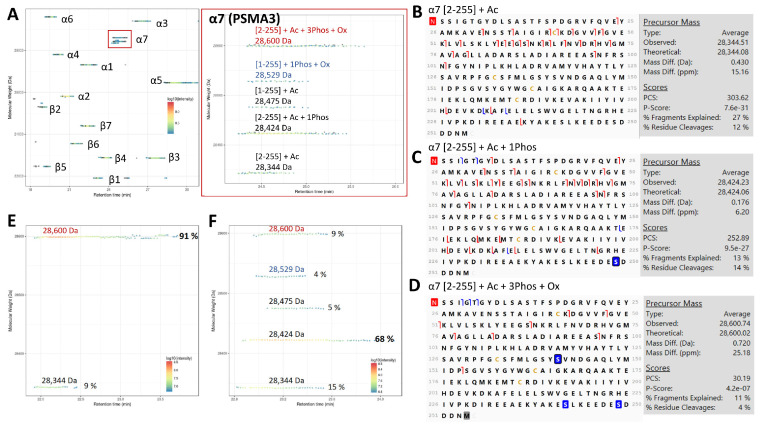
TD-MS identification of proteoforms in human constitutive 20S (c20S) proteasome purified from HEK-EBNA cells: (**A**) Proteoform map visualized in VisioProtMS, showing the c20S subunits zoomed to the different proteoforms of α7 (PSMA3) as shown by the overlapped LC-MS and MS/MS traces. Figures in brackets indicate the first and last amino acids in the sequence of the corresponding detected proteoform. Ac = acetylation, Ox = oxidation, Phos = phosphorylation; (**B**–**D**) MSMS top-down sequencing using Proteome Discoverer of the three main proteoforms of α7 *[2-255]*: acetylated at 28,344 Da, mono-phosphorylated at 28,424 Da and tri-phosphorylated at 28,600 Da. Acetylation, oxidation and phosphorylation sites are highlighted in red, gray, and blue, respectively. Red and blue lines correspond to c-/z- and b-/y- ions, respectively. (**E**,**F**) Changes in the abundances of α7 proteoforms just after purification and after three months of storage at −80 °C, respectively. % correspond to the abundance of the proteoform relative to the sum of intensities corresponding to the subunit.

**Figure 2 cells-12-00844-f002:**
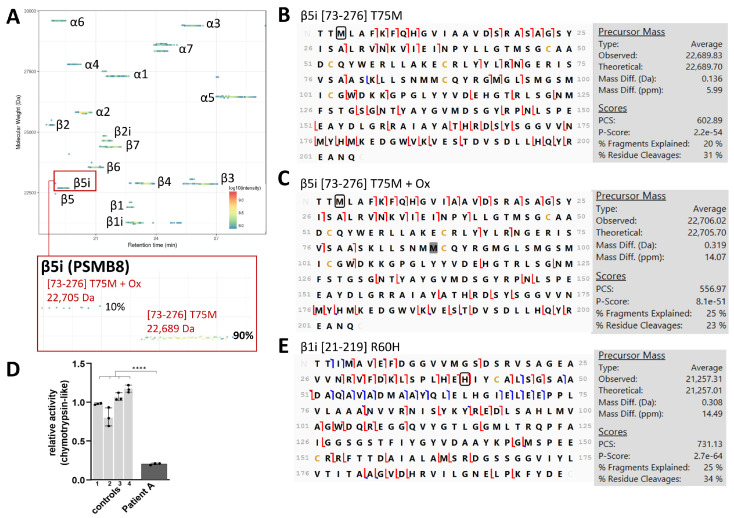
Characterization of the 20S mature proteoforms from PRAAS Patient A with β5i T75M homozygous mutation and β1i R60H SNP by TD-MS: (**A**) Proteoform map using VisioProtMS showing β5i *[73-276]* T75M at 22,689 Da and 22,705 Da (oxidized form). Figures in brackets indicate the first and last amino acids of the mature β5i sequence. Ox = oxidation. % correspond to the abundance of the proteoform relative to the sum of intensity values corresponding to the subunit; (**B**,**C**) MSMS top-down sequencing of the mature *[73-276]* and mature oxidized forms of β5i, respectively. Note that the oxidation, highlighted in gray, was manually localized on one of the methionines in the S153W177 stretch, based on the highest number of *c-* and *z-* fragments. The T75M mutation (circled in black) is clearly confirmed by many *c-* fragments. Red and blue lines correspond to c-/z- and b-/y- ions, respectively; (**D**) Relative chymotrypsin-like activity of the patient compared to healthy controls using Suc-LLVY-AMC substrate analyzed in triplicates (**** *p*-value < 0.0001 from a one-way ANOVA with Tukey’s HSD test); and (**E**) MSMS top-down sequencing of the main R60H mature *[21-219]* β1i proteoform. H60 is circled in black.

**Figure 3 cells-12-00844-f003:**
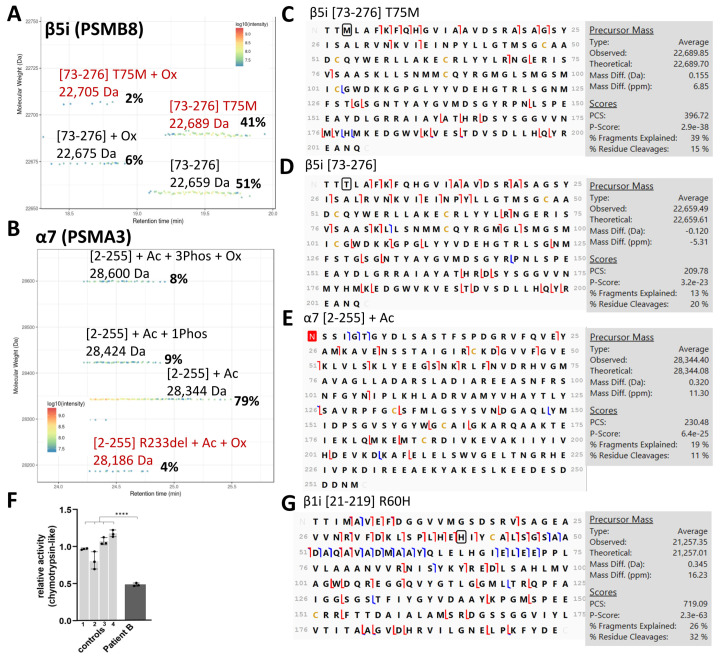
Characterization of the mature 20S proteoforms from PRAAS Patient B with β5i (*PSMB8*) T75M and α7 (*PSMA3*) R233del mutations by TD-MS: (**A**,**B**) Proteoform map using VisioProtMS showing (**A**) both the β5i *[73-276]* wild-type (WT) and T75M at 22,659 Da and 22,689 Da, respectively, and (**B**) α7 *[2-255]* WT (acetylated, monophosphorylated and triphosphorylated forms) and R233del (with acetylation and oxidation) at 28,186 Da. Figures in brackets indicate the first and last amino acids in the sequence of the corresponding detected protein. Ac = acetylation, Ox = oxidation, Phos = phosphorylation. % correspond to the abundance of the proteoform relative to the sum of intensity values corresponding to the subunit; (**C**–**E**) MSMS top-down sequencing using Proteome Discoverer of the (**C**) T75M and (**D**) WT mature *[73-276]* forms of β5i, and the (**E**) acetylated form of α7 (without Met1, acetylation is highlighted in red). Red and blue lines correspond to c-/z- and b-/y- ions, respectively. (**F**) Relative chymotrypsin-like activity of the patient compared to healthy controls using Suc-LLVY-AMC substrate analyzed in triplicates (**** *p*-value < 0.0001 from a one-way ANOVA with Tukey’s HSD test); and (**G**) MSMS top-down sequencing of the main R60H mature *[21-219]* β1i proteoform. H60 is circled in black.

**Figure 4 cells-12-00844-f004:**
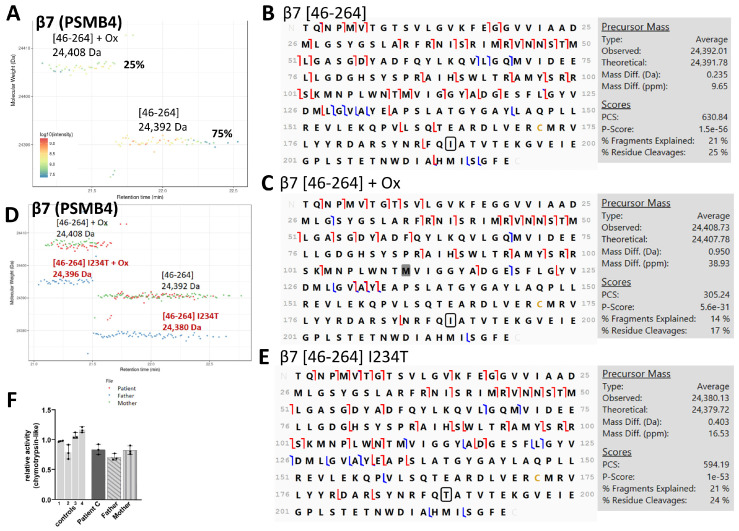
Identification of the mature β7 (*PSMB4*) proteoforms of 20S from PRAAS Patient C with β7 D212_V214del and β7 5’UTR c.-9G>A mutation by TD-MS: (**A**) Proteoform footprint using VisioProtMS showing only the wild-type β7 *[46-264]* mature at 24,392 Da and its oxidized form at 24,408 Da. Ox = oxidation. % corresponds to the abundance of the proteoform relative to the sum of intensity values corresponding to the subunit; (**B**,**C**) MSMS top-down sequencing using Proteome Discoverer of the (**B**) mature *[46-264]* and (**C**) mature plus oxidation proteoforms (note that the oxidation, highlighted in gray, was manually localized on one of the methionines in the middle of the sequence, based on the highest number of c-/z- and b-/y- fragments). Only the mature amino acid sequence of β7 is displayed. Red and blue lines correspond to c-/z- and b-/y- ions, respectively; (**D**) Overlapped proteoform maps of the patient and the parents revealing the −12 Da difference in the major β7 proteoform of the father (24,380 Da vs. 24,392 Da), which may correspond to I234T SNP (circled in black); (**E**) MSMS top-down sequencing of the main proteoform of the father with possible I234T SNP; and (**F**) Relative chymotrypsin-like activity of the patient and the parents compared to healthy controls using Suc-LLVY-AMC substrate analyzed in triplicates.

**Figure 5 cells-12-00844-f005:**
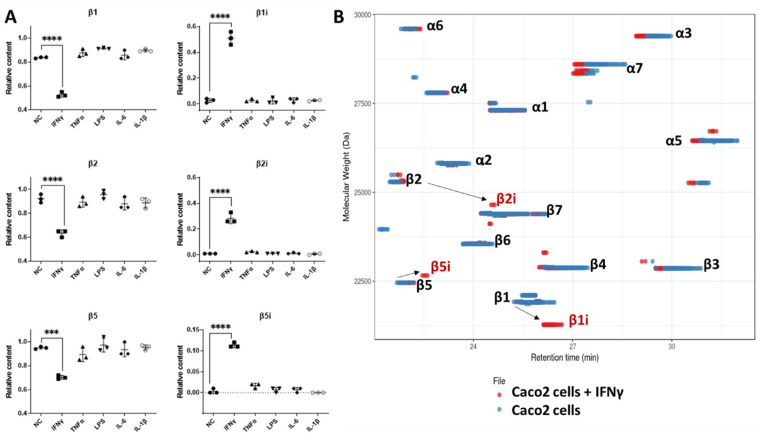
Induction of 20S immuno subunits in Caco2 cells with cytokines: (**A**) Relative amount of c20S and i20S catalytic subunits in the lysate by bottom-up proteomics (intensity-based absolute quantification or iBAQ) upon incubation with different cytokines (10 ng/ml IFNγ, 100 ng/mL TNF-α, 100 ng/mL IL-6, 100 ng/mL IL-1b, 10 ug/mL LPS). (**B**) Overlapped proteoform maps of immunopurified 20S from Caco2 cells with (in red) and without (in blue) IFNγ treatment (10 ng/mL for 7 days of treatment) analyzed with TD-MS showing the induction of immuno subunits. (*** *p*-value < 0.001 and **** *p*-value < 0.0001 from a one-way ANOVA with Tukey’s HSD test).

**Figure 6 cells-12-00844-f006:**
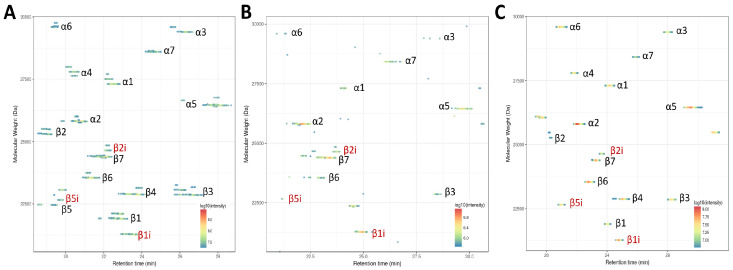
Proteoform map using VisioprotMS showing the TD-MS identification of the immune subunits on immunopurified 20S from different biological samples such as: (**A**) airway organoids derived from normal lung tissue; and (**B**,**C**) biopsies containing 15,000 and 50,000 intestinal crypts, respectively.

**Figure 7 cells-12-00844-f007:**
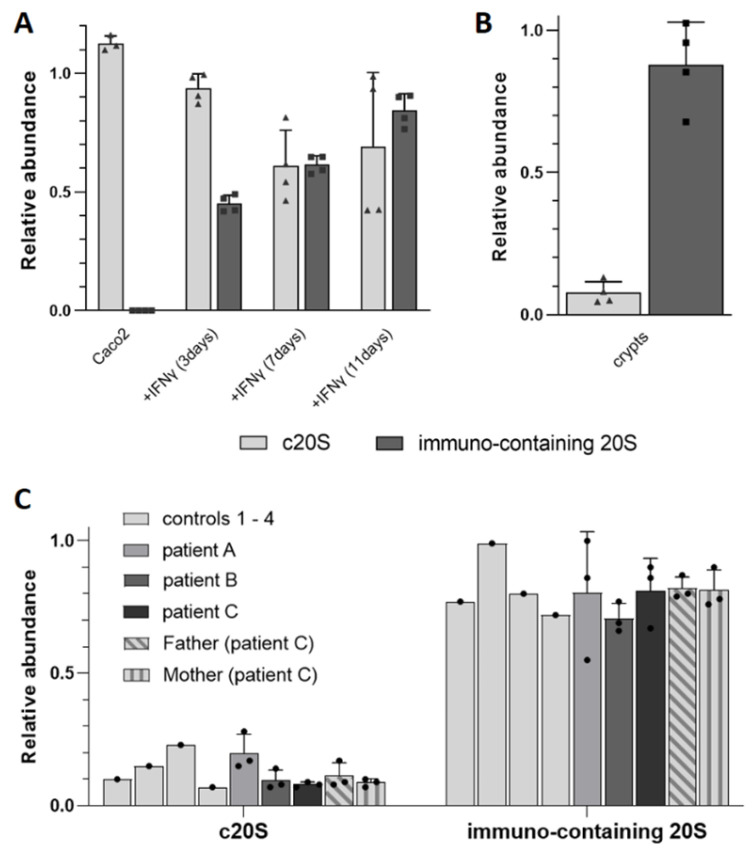
Relative TD label-free quantification of c20S and immuno-containing 20S subunit immunopurified from different samples: (**A**) Caco2 cells (± IFNy), (**B**) intestinal crypts and (**C**) BLCLs derived from PRAAS patients and healthy controls.

**Figure 8 cells-12-00844-f008:**
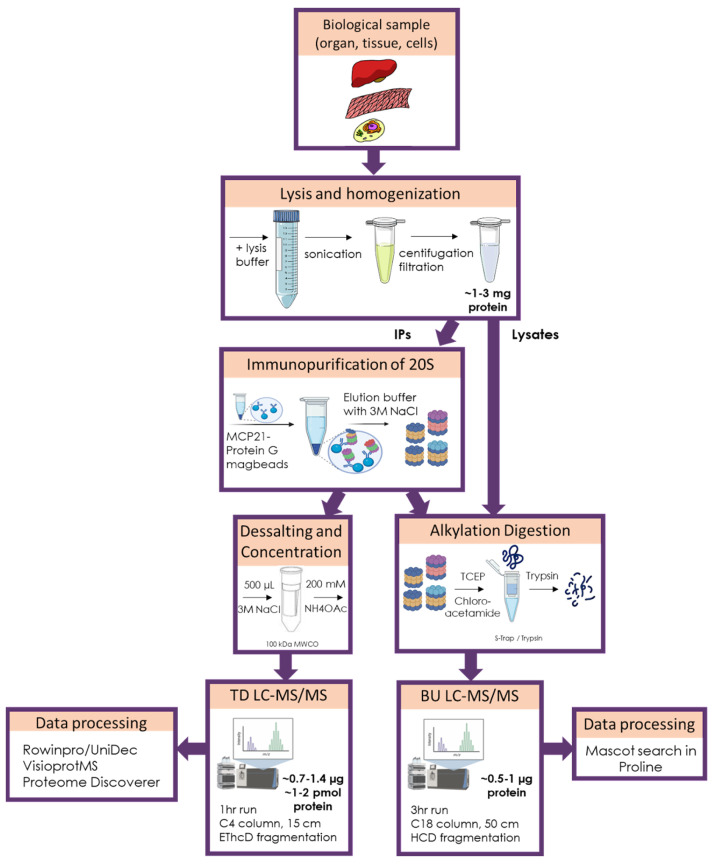
Workflow of our complementary TD-MS and BU-MS analysis of 20S proteasome complexes.

**Table 1 cells-12-00844-t001:** Characteristics of patients from which colonic samples were obtained.

Age (Years)	Sex	Medical Status
66	M	Low-grade pseudoperitoneal myxoma of appendicular origin
59	M	Suspicion of colorectal cancer

## Data Availability

Data is available upon request.

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
