# Peer review of "Establishing 20S Proteasome Genetic, Translational and Post-Translational Status from Precious Biological and Patient Samples with Top-Down MS"

_cells, 2023, doi:10.3390/cells12060844_

Round 1

Reviewer 1 Report

Evaluation of the Manuscript ‘’ Establishing 20S proteasome genetic, translational and post- 2 translational status from precious biological and patient samples with Top-Down MS’’

By Dafun A.S et al

 In this manuscript, the authors develop a rapid TD-MS method to establish the proteoforms of 20S proteasome subunits and to acquire information on their PTMs, mutations related to PRAAS and more common SNPs. Moreover, the authors upgrade the previously established Label-free quantification to precisely and relatively quantify the abundancy of each subunit. Correcting the abundancy by dividing it by the Relative Ionization Yield RIY has enabled more accurate relative quantification of the subunits validated by the BU-MS method. Validation of this new method has occurred at different levels throughout the manuscript indicating the solidity of the method. In addition, the authors present transparently the still existing method limitations, therefore they emphasize that the developed method is currently valid only for the proteasome and needs to be validated specifically for bigger sized proteins or proteins with more complex ionization yields. Overall, I find developing such rapid method to profile the proteasome from different biological samples and which detects PTM and SNPs will bring an advance to the proteasome field and will accelerate the analysis of samples from patients.

My main concern, however, regards the overloading of the manuscript with technical and methodological details on the proteomics workflow and label-free quantification, which makes it very difficult for the non-specialized readership of Cells to understand. I suggest to present the label-free quantification method in a proteomics journal together with the PTM data that could be more detailed and validated. The part from 3.2 contains important information but it is very technology driven and should also be combined with titration of amounts. In this way also the long introduction would be shortened and more focussed.

This manuscript should emphasize the clinical application of the method and the importance of the developed TD-MS method in detecting PRASS mutations and SNPs from different biological samples. The manuscript should better highlight the small amount of sample required for the analysis. It would also be important to obtain specific information on the activity of the immunoproteasome versus the constitutive activities. This would allow the authors to overlay their MS-based characterization with activities for the single active sites, which might provide further insight into the functional consequences of the structural alterations, e.g., due to SNPs or mutations and in clinical biopsy samples. Activity analysis is currently only included by using the three main substrates which do not discriminate between immune- and standard catalytic subunits. Using activity-based probes or specific substrates that are already available and work for purified complexes might add an important layer of information to the TD-MS analysis.

Other concerns:

-        In line 462, correct (Figure S3A, D) to (Figure S3A, B).

-        In line 613 referring to (Figure 5B) is not correct.

-        Line 77, JASL is not described under PSMB10/B2i mutation.

-        Please do not publish the patient identifiers, there is no necessity for that.

-        The authors do not mention the potential of this method to detect other PTM than the presented here.

-        The detection of the PTM presented here is not validated with another method. Therefore, I suggest to validate and bring this into more details in proteomics manuscript as suggested before.

-        The authors claim in line 537-537 that, alpha7 in R233del mutant can be incorporated into the 20s but its expression is very low due to instability. However, they mention that using the BU-MS could not confirm the expression of the alpha7 subunit due to technical limitations. Therefore, they base their conclusion on the fact that no changes in the levels of assembling 20S were observed (measured by quantifying the assembly chaperons) that might indicate a problem with the assembly. I find that the authors should have tried either 1- to use another enzyme than trypsin for the BU-MS validation or 2- to detect the alpha7 subunit in whole lysate and in immunopurified 20s using immuno-blotting. This should give them more specific way of detecting the protein and might detect it in lower molecular weights due to the mutation or instability.

-        Figure 2A: show the composition of all c20s and i20s of the patient with the b5i mutation and not only the b5i.

-        Figure 2B: why the T75M mutation is found at many positions in the presented sequence? Explain more on the sequencing method and how to read the figure legend. Same for figure 3. Please show the deletion inside the sequence. Explain better and have the figures self explanatory.

-        Figure 2 D: Are the three measurements technical replicates or from 2 TD purifications? As you only have 4 healthy controls this seems like a mixing of technical and biological controls which is not appropriate as you skew statistics.  How reproducible is the method? Demonstrate TD accuracy in several biological replicates.

-        Line 476: the 20s interactome was altered by showing the increase of the PA28αβ.

-        Line 560-561: minor correction: ‘’the 5’UTR c.-9G>A 560 mutation’’. It is not a mutation it is SNP.

-        Figure 5B: what are the additional proteoforms that appear in blue and red?

-        For the 15000 and 50,000 Crypts, what are the protein amounts? Please highlight the fact that this method requires little amounts of cells and proteins.

-        Line 631-631: minor correction: ‘’ … to monitor the expression of the i20S…’’. here you monitor the incorporation and not the expression.

-        Figure 6: please add activity data like Activity-based Probes or specific substrate activity to validate your method.

-        Line 635: ‘’response factor’’ that that means?

-        All over the manuscript, please mention always if it was BU-MS or TD-MS.

Reviewer 2 Report

RE: Dafun et. al. Title: “Establishing 20S proteasome genetic, translational and post-translational status from precious biological and patient samples with Top-Down MS”

General comments.

The paper shows an approach to a molecular characterization of the 20S proteo-forms and reveals some new post translational modifications. This raises important questions on what is the cellular role of these modifications and its role in the 20S function. In addition, the authors show this technique could be useful more broadly to define such molecular alterations of other complexes in patient samples and other disease states, which could be useful to better understand these diseases. As initial information or starting point the paper does a good job describing the new modifications and genetic variants associated with diseases. I do not have any major concerns that precludes the acceptance of this manuscript. However, a good faith clarification of details listed below would be important and  would improve the quality of the manuscript:

·         Initially the detection of a new phosphorylation of the co nstitutive 20S is described. However, there was no information presented regarding the status of this proteo-form in i20S, patients or precious samples. If the proteo-form is constitutively presented, then a low trace could be detected in the patients or crypts despite the storage time as depicted in Fig 1E-F.

·         Concern about the statistics applied in different figures of the manuscript: For some of the plots e.g., Supplemental Fig 3, authors present a multiple comparison between patients, A, B, C, and the controls. Due to the nature of the experiment performed, the authors should use a proper multiple comparison test such as ANOVA of multiple t-test with proper correction tests to detect significant results and establish the significance of the experiments. The t-student is solely used just when you compared 2 specific groups and no more (due to error structure for a t-test underestimate the actual error when many groups are being compared). In the case of the multiple bar plots such as supplemental figure 3, figure 5a, figure 10 and figure 8 the natural approach is to perform a multiple comparison to establish the mentioned differences. The authors are advised to make this correction.

·         In line 465, the p.(R60H) SNP with 24% prevalence (0.24) could be considered as a high prevalent allele in population. Moreover, Patient A and B carry the SNP, hence the changes could be due to the additive effect of this mutation plus R233 del or incorporation of the SNP in the 20S which is not mentioned in lines 774-781. Then, the effects measured in patients could be not solely be caused by R233 since there is not enough data to affirm the R233 scenario. In light of this information, could the authors clarify their conclusions here.

·         Authors mentioned no differences in the PIPs and interactome for the patients just PA28ab. However, the authors do not present all the molecules involved in the 20S “interactome". For example, one interactor missing is PA28y, known to be associated with the 20S. So, the statement of no modification in the interactome of the 20S should be corrected. In addition, it is not expected that any PIPs would be seen under the purification conditions that were used. Elution from the MCP21 beads was done with 3M NaCl, it’s well established in the field that concentrations higher than 150mM salt will wash off PIPs. The authors should correct there discussion in light of this information, and it would not be rigorous to discus PIP differences with such a stringent purification condition.

·         The authors also do not acknowledge the fact that many of the 20S they  isolate from the MCP21 beads likely come from 26S  complexes that have fallen apart due to lack of ATP in isolation conditions, and high salt. This perspective is an important qualifier so the reader is aware that the 20S that is being quantified, could have been a 26S proteasome in the cell.

·         In line 612 healthy organoids cite figure 5B, but panel 5B is label as caco2 cells. I believe, the correct label for line 612 is figure 6B.

·         In the RIY item 3.2 line 634:656, explanation of the normalization procedure will benefit if the authors add the equations to visually understand this normalization. The explanation is somehow confusing and did not depict the exact procedures used for the operations, the addition of a graphical equation will improve the RIY quantification/normalization.

·         How did the authors test the reproducibility of this normalization? Do the authors check random datasets and compare the performance or did the authors have any ROC curve to assess the performance of RIY normalization and to compare the performance of both approaches NGC and RIY? As suggestion for the authors, adding ROC will complement the results and increase the value and robustness to the normalization. It will also show the sensitivity of this normalization against false positives and false negatives.

·         In figure 10A-C. What statistic was used in the comparison? was ANOVA or t-test? This should be clarified in the text and figure legends.

·         For Patient A there is a decrease trend in the caspase like activity not significant but still represents a reduction figure S3. However, the authors stated that it had not changed in line 779-780. As you can depict from figure s3s there is indeed a reduction in the activity so the claim should be corrected or clarified.

·         The authors mentioned the sensitivity in pmol of this TD approach. When compared with BU MS, what was the amount of sample used?

·         The authors mentioned the expected ratios in line 809, were the values expected results from a ROC model, logistic model? There are no expected ratios supplied in the manuscript or supplemental data, so, where do these expected ratios come from? This should be defined in the manuscript.

·         Finally, as a suggestion and due to the nature of the protocol the authors will benefit from the addition of scheme workflow/flowchart for this fast approach depicting the major aspects of the method and supplement the equations for the normalization process.

In general, the conclusions highlighted by the authors make this manuscript sufficient for publication in this journal; the description and the method presented to quantify the ratios are supported by the data and results. The newly found PTMs enrich the knowledge of proteo-form diversity in the 20S proteasome and open the possibilities for new types of regulation of the 20S. The question remains how these PTMs could affect the function/structure of the proteasome and whether they are essential for its proper functioning in different diseases.

Regarding the c20S/i20S ratios, the paper shows a remarkably interesting approach to determine the i20S ratios and the clinical use of this technique could make a difference when quickly characterizing different immune responses in precious samples such as biopsies. The RIY normalization model for the different types of MS still has room for refinement, but it does present a step towards a better understanding and use of MS data is sensitive cases such as immune response and the switch from c20S to i20S. Overall, this article moves step towards the correct detection, interpretation of PTMs and point mutations impact in the function of fully folded proteins and protein complexes, which has a great contribution to biology and of course to the understanding of the 20S proteasome biology.

Round 2

Reviewer 1 Report

I thank the authors for the comprehensive revision of the manuscript and addressing my concerns. Most of them were appropriately addressed only two minor ones need to be resolved.  

Minor concerns:

1. Lines 73ff: “PRAAS spans the disease spectrum ranging from chronic atypical neutrophilic dermatosis with lipodystrophy and elevated temperature (CANDLE), joint contractures, muscle atrophy, mi-crocytic anemia and panniculitis-induced lipodystrophy (JMP), Nakajo-Nishimura syndrome (NNS) and Japanese autoinflammatory syndrome with lipodystrophy (JASL) which are autoinflammatory syndromes caused by genetic mutations in the immunoproteasome subunits, β5i encoded by PSMB8, and in β2i (PSMB10)… “

The aforementioned PRAAS syndromes relate mainly to mutations in beta51 and beta 1i (PSMB9). There is only one PSMB10 mutation described so far, which has a mild phenotype (doi: 10.1016/j.jaci.2019.11.024). Please correct!

2.Please make sure that all legends contain a clear reference to Top down or bottom up methods used.

Author Response

We are grateful to both reviewers for their very careful and thoughtful review.

Please see the attachment for our point-by-point response to the minor concerns raised by reviewer 1.

Reviewer 2 Report

The responses from the authors have been carefully reviewed. I found that the authors have adequately addressed the raised concerns by making appropriate corrections in the manuscript and by providing the necessary clarifications. The additional information requested, such as the equations for the normalization process, the comparison with BU MS in terms of sample amounts, the implementation of correct statistics and further explanation on the expected ratios of 20S were provided in a clear and concise manner.

Based on the new revised manuscript, I believe that the research presented is significant and of quality for this journal. The authors have provided a thorough description of their experimental design, results, and interpretation of the data. Additionally, the methodology used in the study is appropriate and represent a novel approach to detect new modifications in the 20S proteasome.

Overall, I find that the manuscript is suitable for publication in the journal and I recommend that it be accepted without further revisions.

Author Response

We are grateful to both reviewers for their very careful and thoughtful review.